# MGMT Promoter Methylation and IDH1 Mutations Do Not Affect [^18^F]FDOPA Uptake in Primary Brain Tumors

**DOI:** 10.3390/ijms21207598

**Published:** 2020-10-14

**Authors:** Andrea Cimini, Agostino Chiaravalloti, Maria Ricci, Veronica Villani, Gianluca Vanni, Orazio Schillaci

**Affiliations:** 1Department of Biomedicine and Prevention, University Tor Vergata, 00133 Rome, Italy; agostino.chiaravalloti@gmail.com (A.C.); maria.ricci28@gmail.com (M.R.); orazio.schillaci@uniroma2.it (O.S.); 2Nuclear Medicine Section, IRCCS Neuromed, 86077 Pozzilli, Italy; 3Neuro-Oncology Unit, Regina Elena National Cancer Institute, 00144 Rome, Italy; veronicavillani79@gmail.com; 4Department of Surgical Science, Tor Vergata (PTV) University, 00133 Rome, Italy; Vanni_gianluca@yahoo.it

**Keywords:** Primary brain tumors, nuclear medicine, positron emission tomography, [^18^F]FDOPA, MGMT promoter methylation, IDH1 mutation, radiopharmaceuticals

## Abstract

The aim of our study was to investigate the effects of methylation of *O⁶-methylguanine-DNA methyltransferase promoter* (MGMTp) and isocitrate dehydrogenase 1 (IDH 1) mutations on amino acid metabolism evaluated with 3,4-dihydroxy-6-[^18^F]-fluoro-l-phenylalanine ([^18^F] FDOPA) positron emission tomography/computed tomography (PET/CT). Seventy-two patients with primary brain tumors were enrolled in the study (33 women and 39 men; mean age 44 ± 12 years old). All of them were subjected to PET/CT examination after surgical treatment. Of them, 29 (40.3%) were affected by grade II glioma and 43 (59.7%) by grade III. PET/CT was scored as positive or negative and standardized uptake value ratio (SUVr) was calculated as the ratio between SUVmax of the lesion vs. that of the background. Statistical analysis was performed with the Mann–Whitney U test. Methylation of MGMTp was detectable in 61 out of the 72 patients examinated. Mean SUVr in patients without methylation of MGMTp was 1.44 ± 0.38 vs. 1.35 ± 0.48 of patients with methylation (*p* = 0.15). Data on IDH1 mutations were available for 43 subjects; of them, 31 are IDH-mutant. Mean SUVr was 1.38 ± 0.51 in patients IDH mutant and 1.46 ± 0.56 in patients IDH wild type. MGMTp methylation and IDH1 mutations do not affect [^18^F] FDOPA uptake in primary brain tumors and therefore cannot be assessed or predicted by radiopharmaceutical uptake parameters.

## 1. Introduction

In the recent World Health Organization (WHO) Classification of Tumors of the Central Nervous System (2016), molecular genetic alterations have been incorporated to the classic histology of primary brain tumors (PBT) [1], including isocitrate dehydrogenase 1 and 2 (IDH1 and IDH2) mutations and methylation of the *O⁶-methylguanine-DNA methyltransferase promoter* (MGMTp). The IDH mutation is associated with a better prognosis in patients with glioma, independently of histological parameters and tumor grade [2]. 

MGMTp methylation silences the MGMT gene and reduces the ability of tumor cells to repair damage caused by temozolomide and other alkylating agents [1]. Moreover, MGMTp methylation confers a better prognosis to high grade gliomas [3,4].

Positron emission tomography (PET) with amino acid tracers ^11^C-methyl-l-methionine ([^11^C]MET), O-(2-[^18^F]fluoroethyl)-l-tyrosine ([^18^F]FET), and 3,4-dihydroxy-6-[^18^F]-fluoro-l-phenylalanine ([^18^F]FDOPA) is widely used in the management of gliomas for tumor grading, differential diagnosis, delineation of tumor extent, surgical and radiotherapy treatment planning, and post-treatment surveillance [5]. These radiolabeled compounds allow the visualization of the amino acid radiopharmaceuticals uptake mediated by the L-type amino acid transporter 1 and 2 (LAT1 and LAT2) and related to the augmented protein synthesis in brain tumor cells; nevertheless, the relationship between amino acid tracers uptake and molecular parameters in the context of the recent WHO classification is still unclear. 

A previous paper by Lopci et al. evaluated the count rate (the number of counts per second measured by the scanner) in a ROI drawn on the area of the tumor with the highest uptake of [^11^C] MET in patients with supratentorial glioma who underwent surgery. The study mainly explored the relationship between this parameter and clinical biological data (including IDH1 mutation status, 1p/19q codeletion and MGMT promoter methylation), describing a significant correlation with histological grade and IDH1 mutation status [6]. Moreover, previous data suggested [^18^F] FET PET imaging may allow a non-invasive evaluation of IDH mutation status in gliomas [7,8], even if IDH mutated and 1p/19q co-deleted oligodendrogliomas cannot be differentiated from glioblastomas and astrocytomas by [^18^F] FET PET [7].

In 2017, the results of the study of Verger et al. evaluating 43 patients with grade II and III gliomas, conversely showed higher [^18^F]FDOPA uptake in gliomas with IDH mutation [9]; nevertheless, in the study of Cicone et al. [10], including 33 patients with glioma (grade II, III, and IV gliomas), [^18^F]FDOPA uptake parameters did not show any correlation with IDH status, with no significant differences between IDH mutant and IDH wild-type gliomas.

Regarding the MGMTp methylation, Okita et al. found a significant correlation between MET uptake and MGMTp methylation in patients with non-enhancing grade II and III glioma [11]. In contrast, other authors have not found any correlation between amino acid tracers and MGMTp methylation status [6,12].

The aspects concerning the possible correlation between imaging and genetic biomarkers may lead to novel insight into glioma physiopathology, potentially improving its management.

Nevertheless, to date, the scientific literature described heterogeneous results, and therefore, further studies are needed in order to determinate the possible relationship between amino acid radiopharmaceuticals metabolism and molecular genetic parameters. In fact, to the best of our knowledge, the relationship between molecular alterations and uptake of [^18^F]FDOPA, an analog of l-3,4-dihydroxyphenylalanine (L-DOPA) with a similar metabolic pathway in cells, has been examined only in two reports [9,10] with contrasting results. Therefore, this study aims to investigate the effects of MGMTp methylation and IDH1 mutations on amino acid metabolism with [^18^F]FDOPA PET/computed tomography (CT) in patients with glioma.

An initial version of this paper has been presented as a conference paper at the 32nd Annual Congress of the European Association of Nuclear Medicine. 

## 2. Materials and Methods

### 2.1. Patients

Seventy-two patients with primary brain tumors were included in this retrospective study (33 women and 39 men; mean age 44 ± 12 years old). After surgical treatment, all patients were subjected to PET/CT examination with [^18^F]FDOPA at Policlinico Tor Vergata (Rome, Italy) or at IRCCS Neuromed (Pozzilli, Italy) between December 2011 and March 2019: the time from surgery to imaging ranges from six months to 41 months (median eight months) in the study population. The subjects included in the retrospective analysis were chosen according to the following criteria: adult age (≥18 years old), the availability of brain tumor molecular genomic information (IDH1 status and/or MGMTp methylation, detected respectively by immunohistochemistry and methylation specific PCR) and willingness to participate in the present study as demonstrated by providing written informed consent. Patients with glioblastoma (grade IV glioma) were excluded from the study for the low number of patients available (only two). Moreover, the MGMTp methylation status in these tumors may change after surgery, representing a possible bias for our analysis [13,14]. The study was performed according to the declaration of Helsinki [15]. 

Of the 72 patients, 37 patients were affected by astrocytoma (51.4%), three by anaplastic astrocytoma (4.1%), two by oligoastrocytoma (2.8 %), two by brain stem glioma (2.8 %), and 28 by oligodendroglioma (38.9%). Regarding tumor grade, 29 patients were affected by grade II glioma (40.3 %) and 43 (59.7%) by grade III glioma. 

### 2.2. [18F]FDOPA PET/CT, SUVmax, SUVratio, and Image Evaluation

PET/CT images were acquired using a Discovery VCT or a Discovery ST 16 scanner (GE Healthcare, Chicago, Illinois, USA) 20 min after [^18^F]FDOPA injection (180 ± 75 MegaBequerels). No carbidopa was administered before radiotracer injection. PET/CT acquisition lasted 12 min in all patients. Image reconstruction was performed using ordered-subsets expectation maximization (OSEM) with 20 subsets and ordered-subsets expectation maximization (OSEM) with 20 subsets and four iterations. Attenuation correction with a low-ampere CT scan of the head (40 mA; 120 kV) was performed before PET image acquisition. Figure 1 shows two lesions of the study with high uptake of [^18^F]FDOPA.

A volume of interest (VOI) on the recurrence site was traced by an experienced nuclear medicine physician (A.Ch.) starting from the slice with the highest uptake of the radiopharmaceutical. Moreover, the use of co-registered MRI images [16] permitted a correct placement of the VOI, even in tumors with involvement of the striatum (Figure 1A). From the VOI, the maximum standardized uptake value for the site of recurrence (SUVmax lesion) was calculated on a dedicated workstation (version 4.4, Advantage Workstation, GE Healthcare, Chicago, IL, USA). No one of the patients of this study had PBT in the occipital region; therefore, this site was chosen for SUVmax calculation for the background (SUVmax occ), obtained using a standard VOI of 1.5 cm × 1.5 cm ×1.5 cm, placed on the occipital lobe as proposed in a previous study of Chiaravalloti et al. [17].

SUVratio (SUVr) was calculated as SUVmax/SUVmax occ [16].

### 2.3. Statistical Analysis

We calculated the means and the standard deviation of the results of semiquantitative analysis for SUVr in different molecular genetic parameters. Possible differences in SUVr values for patients with different molecular genetic parameters (MGMTp methylation vs. no MGMTp methylation; IDH1 mutant vs. IDH1 wildtype) have been assessed using the Mann–Whitney U test.

### 2.4. Ethics Statement

This research was approved by Comitato Etico Istituto Neurologico Mediterrraneo Neuromed (Chiaravalloti2018/2021, 01/12/2018). Written informed consent was obtained from each of the donors.

## 3. Results

Methylation of MGMTp was detectable in 61 out of the 72 patients examinated (84.7%). Data on IDH1 mutations were available for 43 subjects; 31 were IDH1 mutant (72.1%). According to the 2016 WHO classification of Tumors of the Central Nervous System: 26 patients had grade II glioma and MGMTp methylation (36.1%); three patients had grade II glioma but no MGMTp methylation (4.2%); 35 patients had grade III glioma and MGMTp methylation (48.6%); eight patients had grade III glioma but no MGMTp methylation (11.1%); 14 patients had grade II glioma IDH1 mutant (32.5%); four patients had grade II glioma but no mutation of IDH 1 (IDH1 wild-type) (9.3%); 17 patients had grade III glioma IDH1 mutant (39.5%); and eight patients had grade III glioma IDH wild-type (18.6%). A general overview of study population with the molecular genetic parameters is reported in Table 1.

The Mann–Whitney U test showed no statistical difference (*p* = 0.15) in [^18^F]FDOPA uptake between patients with methylation of MGMTp and patients with no methylation of MGMTp (mean SUVr 1.44 ± 0.38 vs. 1.35 ± 0.48 respectively). Moreover, no significant difference (*p* = 0.79) in [^18^F]FDOPA uptake has been demonstrated between patients IDH1 mutant and IDH1 wild type (mean SUVr 1.38 ± 0.51 vs. 1.46 ± 0.56, respectively). An overview of the final results is provided in Table 2 and in Figure 2.

## 4. Discussion

In light of the new WHO molecular classification for brain tumors (2016), the possible influence of genomic markers on the degree of amino acid radiopharmaceuticals uptake has been discussed [18]. The role of amino acid radiopharmaceuticals in tumor grading and patient prognosis is well confirmed. In contrast, their correlation with molecular markers is still to be established: the few studies conducted have led to a heterogeneity of results, and the overall scenario in the scientific literature is not yet clear. The potential of PET with amino acid radiopharmaceuticals for the prediction of genomic alterations in PBT is still under discussion.

To the best of our knowledge, this is the first study to correlate the methylation status of MGMT promoter with the degree of [^18^F]FDOPA uptake in patients with glioma, showing no significant differences between the two patient groups examined (*p* = 0.15). Our results and the findings of the studies of Lopci et al. [6] and Ribom et al. [12] with [^11^C]MET PET highlight an overall lack of correlation between the degree of amino acid tracers uptake and the methylation status of MGMT promoter: in comparison with the previous reports mentioned above, we used a three-dimensional VOI for the assessment of semiquantitative PET parameters to evaluate better a possible heterogeneity of amino acid metabolism in the tumor.

As regards the IDH1 status, no statistically significant differences were found in the degree of [^18^F]FDOPA uptake between IDH1 mutant and IDH1 wildtype patients (*p* = 0.79), according to the results of the study by Cicone et al., with a slightly higher number of patients in our analysis (43 vs. 31) [10]; our results are also in contrast with the previous report of Verger et al. [9], in which a paradoxically higher [^18^F]FDOPA uptake in IDH mutant glioma was shown. Nevertheless, the authors’ methodology was different (bidimensional ROI used by Verger et al. [7] vs. tridimensional VOI used in our analysis), that may explain the contrasting results. 

Considering our findings with those of previous studies with [^11^C]MET and [18F]FET PET [6,7,8], in which higher values of uptake of the radiopharmaceutical in IDH wild-type gliomas compared to IDH mutant gliomas were demonstrated, we support the hypothesis of different behavior of the IDH status of glioma on amino acid tracers metabolism and in particular a lack of influence only on [^18^F]FDOPA uptake. Nevertheless, it is essential to underline that the heterogeneity of results in literature may depend on different study methods or bias in selection of patients as well; hence, further studies are needed to confirm the lack of correlation between [^18^F]FDOPA uptake and the IDH status of gliomas.

Regarding the semiquantitative values of the analysis, the normalization of SUV (i.e., ratio between pathological and healthy tissue) represented the best strategy to evaluate PET exams with [^18^F]FDOPA in this study since SUV values may vary significantly using different injected activities, tomographs, or reconstruction parameters [19].

The study had some limitations: first, the results are obtained retrospectively; second, the much greater number of patients with MGMTp methylation in comparison to patients with no MGMTp methylation (61 vs. 11 respectively) and the more significant number of patients with IDH1 mutation in comparison to patients IDH1 wild-type (31 vs. 12 respectively); future studies with numerically balanced subgroups are needed in order to confirm our findings. Furthermore, another significant aspect to consider is the long time interval between the date of surgery and date of execution of PET/CT examination with [^18^F]FDOPA (> 1 year in many cases). During this period, it is not possible to exclude a switch of low-grade gliomas toward more malignant forms, representing a bias for our analysis. In a recent study, Murphy et al. [20] demonstrated a conversion rate of low-grade gliomas to more aggressive forms of 21%, with a median transformation time of 56 months. Moreover, as mentioned above, we excluded the most aggressive forms (grade IV gliomas) from our analysis: our results cannot be applied to all types of gliomas.

## 5. Conclusions

Based on these results, the molecular-genomic characteristics of gliomas are not correlated with the degree of [^18^F]FDOPA uptake and therefore cannot be assessed or predicted by radiopharmaceutical uptake parameters.

## Figures and Tables

**Figure 1 ijms-21-07598-f001:**
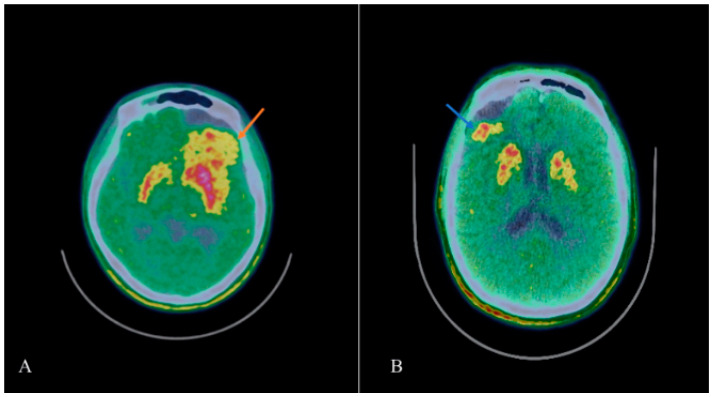
(**A**) [^18^F]FDOPA PET/CT image of patient #3 (III grade oligondendroglioma, *O⁶-methylguanine-DNA methyltransferase promoter* (MGMTp) methylated, and isocitrate dehydrogenase (IDH) mutant) demonstrates an area of pathologic accumulation of the tracer in the left frontal lobe (orange arrow, SUVmax 3.2) with the involvement of the ipsilateral striatum. (**B**) [^18^F]FDOPA PET/CT image of patient #27, II grade astrocytoma, MGMTp methylated, and IDH1 wild type shows an area of pathological uptake of the radiopharmaceutical (blue arrow, SUVmax 2.9) in the posterior region of the surgical cavity, localized in the right frontal lobe.

**Figure 2 ijms-21-07598-f002:**
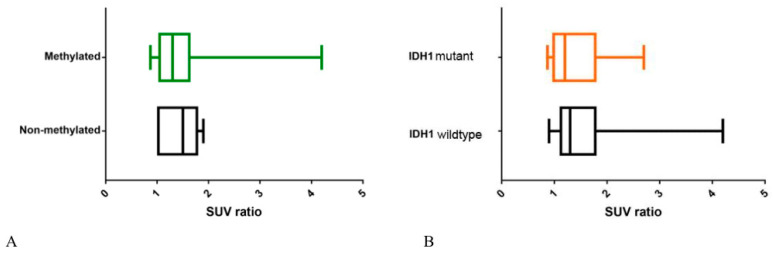
Box-plots showing no statistical differences in (**A**) [^18^F]FDOPA uptake between patients with methylation of *O⁶-methylguanine-DNA methyltransferase promoter* (MGMTp) and patients with no methylation of MGMTp (*p* = 0.15) and (**B**) between patients isocitrate dehydrogenase 1 (IDH1) mutant and IDH1 wild-type (*p* = 0.79).

**Table 1 ijms-21-07598-t001:** Overview of study population with the molecular genetic parameters.

Patient	Tumor Type	Tumor Grade	*O⁶-methylguanine-DNA methyltransferase promoter* (MGMTp) Methylation (Y/N)	Isocitrate Dehydrogenase 1 (IDH1) Mutation (Y/N)	SUVma × Lesion	SUVma × occ	SUVr
#1	Astrocytoma	2	Y		0.9	0.9	1
#2	Astrocytoma	3	Y		1.6	1.6	1
#3	Oligodendroglioma	3	Y	Y	3.2	1.3	2.46
#4	Oligodendroglioma	3	Y	Y	3.4	2.2	1.54
#5	Oligodendroglioma	3	Y		3.4	1.9	1.79
#6	Anaplastic Astrocytoma	3	Y		0.9	0.8	1.12
#7	Astrocytoma	2	Y	Y	0.9	0.7	1.28
#8	Anaplastic Astrocytoma	3	Y	Y	1.2	0.9	1.33
#9	Astrocytoma	2	N	Y	2.1	1.2	1.75
#10	Brain Stem Glioma	2	Y		1.7	1.2	1.42
#11	Astrocytoma	2	N	Y	2.1	1.2	1.75
#12	Brain Stem Glioma	2	Y		1.7	1.2	1.42
#13	Astrocytoma	3	Y		1.9	1.5	1.26
#14	Oligodendroglioma	2	Y		0.8	0.9	0.88
#15	Astrocytoma	3	N		2.6	1.6	1.62
#16	Astrocytoma	3	N		2.6	1.4	1.86
#17	Oligodendroglioma	2	Y	Y	1.7	1.4	1.21
#18	Oligodendroglioma	2	Y	Y	1.5	1.3	1.15
#19	Oligodendroglioma	2	Y		1.5	1	1.5
#20	Astrocytoma	2	Y	N	1.1	0.9	1.22
#21	Oligoastrocytoma	2	Y		1.4	1.5	0.93
#22	Astrocytoma	2	Y		1.4	1.0	1.4
#23	Astrocytoma	3	Y		1.7	1.1	1.54
#24	Oligoastrocytoma	2	Y	Y	2.2	1.4	1.57
#25	Oligodendroglioma	3	Y	Y	0.8	0.9	0.89
#26	Oligodendroglioma	3	Y		1.5	1.5	1
#27	Astrocytoma	2	Y	N	2.9	1.5	1.93
#28	Oligodendroglioma	3	Y	Y	0.94	0.72	1.3
#29	Astrocytoma	3	Y		2.09	1.05	1.99
#30	Oligodendroglioma	2	Y	Y	1.2	0.9	1.33
#31	Oligodendroglioma	2	Y		2	1.6	1.25
#32	Astrocytoma	3	Y	N	2.3	1	2.3
#33	Astrocytoma	3	Y		2	1	2
#34	Oligodendroglioma	2	Y		1.3	0.8	1.62
#35	Astrocytoma	3	N	Y	2	1.4	1.43
#36	Astrocytoma	3	Y	Y	1.2	0.9	1.33
#37	Astrocytoma	3	Y		2.4	1.3	1.85
#38	Astrocytoma	2	Y		1.1	1	1.1
#39	Astrocytoma	2	Y		1.6	1.5	1.07
#40	Oligodendroglioma	2	Y	Y	5.2	1.8	2.89
#41	Astrocytoma	3	Y	Y	0.9	1	0.9
#42	Astrocytoma	3	Y	Y	1.5	1.1	1.36
#43	Oligodendroglioma	3	Y	Y	1.2	1.1	1.09
#44	Astrocytoma	2	Y	Y	1.2	1.4	0.86
#45	Anaplastic Astrocytoma	3	Y	Y	1.7	1.3	1.31
#46	Oligodendroglioma	3	Y	Y	1.5	0.8	1.87
#47	Oligodendroglioma	3	Y		1.6	1.1	1.45
#48	Oligodendroglioma	2	Y	Y	1.6	1.4	1.14
#49	Oligodendroglioma	3	Y	Y	3.04	2.17	1.4
#50	Oligodendroglioma	3	Y		2.1	2.04	1.03
#51	Oligodendroglioma	3	Y		2.1	2.07	1.01
#52	Astrocytoma	3	Y	Y	8.2	1.95	4.2
#53	Astrocytoma	3	Y		1.57	1.54	1.02
#54	Astrocytoma	3	Y		2.51	1.41	1.78
#55	Astrocytoma	3	Y		3.58	1.96	1.83
#56	Astrocytoma	3	Y		3.43	2.02	1.7
#57	Astrocytoma	3	Y	Y	2.88	2.21	1.3
#58	Astrocytoma	3	N	N	5.13	4.6	1.11
#59	Astrocytoma	3	N	N	2.34	2.3	1.02
#60	Astrocytoma	3	Y	N	7.07	2.65	2.67
#61	Astrocytoma	3	Y	N	2.85	1.56	1.83
#62	Astrocytoma	3	Y	N	3.58	2.04	1.75
#63	Astrocytoma	3	Y	N	2.72	1.73	1.57
#64	Oligodendroglioma	3	N	Y	2.41	2.43	0.99
#65	Oligodendroglioma	3	N	Y	3	3.03	0.99
#66	Oligodendroglioma	2	Y	N	2.88	2.97	0.96
#67	Oligodendroglioma	2	Y	N	1.94	2.22	0.87
#68	Oligodendroglioma	2	Y	Y	3.77	2.68	1.4
#69	Astrocytoma	3	N	N	2.18	2.24	0.89
#70	Oligodendroglioma	2	Y	Y	2.81	2.36	1.19
#71	Astrocytoma	2	N	Y	2.51	2.67	0.94
#72	Oligodendroglioma	2	Y	Y	1.8	1.7	1.05

**Table 2 ijms-21-07598-t002:** General overview of final results.

Patients	Mean SUVr ± Standard Device	*p* (Mann Whitney U Test)
*O⁶-methylguanine-DNA methyltransferase promoter* (MGMTp) methylation vs. no MGMTp methylation	1.44 ± 0.38 vs. 1.35 ± 0.48	0.15
Isocitrate dehydrogenase 1 (IDH1) mutant vs. IDH1 wild-type	1.438 ± 0.51 vs. 1.46 ± 0.56	0.79

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
