# Peer review of "MGMT Promoter Methylation and IDH1 Mutations Do Not Affect [18F]FDOPA Uptake in Primary Brain Tumors"

_ijms, 2020, doi:10.3390/ijms21207598_

Round 1
Reviewer 1 Report
The authors present the paper “MGMT promoter methylation and IDH1 mutations do not affect [18F]FDOPA uptake in primary brain tumors.” In this article, the authors investigated whether methylation (MGMTp) or IDH1 effects amino acid metabolism as evaluated with [18F] FDOPA. They found that there is no correlation, and therefore this cannot be used as a noninvasive method to identify molecular alterations in tumors. Although this is negative data, the authors use this study to help clarify conflicting data in the literature. My comments to the authors are as follows:
Introduction:The introduction pulls in a lot of the relevant literature on this topic, but some of the sentences are not clear, making it difficult to read (in particular line 52-60 and 70-76) and understand the author’s point.
Materials and methods: Line 93- this should start with the word “Of” instead of “Off.”
Figure 1- Although the images are clear, it would be helpful if the authors included arrows or a different type of marker to identify the region of increased SUV for readers who are not familiar with reading PET scans
Results: As the authors point out, this method has been used to identify tumor grade/prognosis. It would be helpful if the authors included this correlation in their study to show that the identified cohort is representative of what has already been published in literature.
Conclusion: There are more patients with MGMT methylation (61 vs 11) and more patients with IDH1 mutation (31 vs 12) in the cohort studied, which would select for a patient population that is more likely to have a better prognosis. It would be helpful if the authors included information about the number of patients needed to find a statistical difference between the two groups to confirm that the negative findings are truly due to no difference, and not because there weren’t enough patients analyzed in the MGMT non-methylated or IDH1 wt groups to find a statistical difference. In other words, the authors should include information about what n is needed for statistical significance to ensure that this was reached for both groups.
The authors note that the time to analysis is a limitation because tumors may have transformed from a low grade tumor to a high grade tumor within this time frame, which may alter the results. It is also very important to acknowledge that patients who live longer by default have a better prognosis. The authors seem to be selecting for a cohort that is less likely to have aggressive features- which also will alter results, but in an opposite manner as presented by the authors already in the conclusion.
Author Response
Point1: Introduction:The introduction pulls in a lot of the relevant literature on this topic, but some of the sentences are not clear, making it difficult to read (in particular line 52-60 and 70-76) and understand the author’s point.
Response 1: As kindly suggested, we have modified some sentences (especially line 52-60 and 70-76) in order to make more comprehensible the introduction.
Point2: Materials and methods: Line 93- this should start with the word “Of” instead of “Off.”
Response2: As kindly suggested, we have changed “Off” to “Of”.
Point3: Figure 1- Although the images are clear, it would be helpful if the authors included arrows or a different type of marker to identify the region of increased SUV for readers who are not familiar with reading PET scans.
Response3: As kindly suggested, we have included arrows in figure 1.
Point4: Results: As the authors point out, this method has been used to identify tumor grade/prognosis. It would be helpful if the authors included this correlation in their study to show that the identified cohort is representative of what has already been published in literature.
Response4: Although it would be very helpful, unfortunately we haven’t sufficient data for analyze this correlation. The main aim of this study was to correlate molecular mutations to [18F]FDOPA uptake, and therefore to analyze if MGMTp methylation status and IDH1 mutation status could be predicted by [18F]FDOPA uptake parameters: our data collection was mainly focused on nuclear medicine parameters and mutation information.
Point5: Conclusion: There are more patients with MGMT methylation (61 vs 11) and more patients with IDH1 mutation (31 vs 12) in the cohort studied, which would select for a patient population that is more likely to have a better prognosis. It would be helpful if the authors included information about the number of patients needed to find a statistical difference between the two groups to confirm that the negative findings are truly due to no difference, and not because there weren’t enough patients analyzed in the MGMT non-methylated or IDH1 wt groups to find a statistical difference. In other words, the authors should include information about what n is needed for statistical significance to ensure that this was reached for both groups.
Response5: Although the Mann Whitney U test is optimized for size greater than 7 patients (https://www.graphpad.com/guides/prism/8/statistics/how_the_mann-whitney_test_works.htm), our data are not well balanced. As kindly suggested, we have included a sentence in the discussion regarding information about the number of patients, specifying that future studies with well-balanced molecular subgroups are needed: “The study had some limitations: first, the results are obtained retrospectively; second, the much greater number of patients with MGMTp methylation in comparison to patients with no MGMTp methylation (61 vs 11 respectively) and the greater number of patients with IDH1 mutation in comparison to patients IDH1 wildtype (31 vs 12 respectively): future studies with numerically balanced subgroups are needed in order to confirm our findings” (L 197-201).
Point6: The authors note that the time to analysis is a limitation because tumors may have transformed from a low grade tumor to a high grade tumor within this time frame, which may alter the results. It is also very important to acknowledge that patients who live longer by default have a better prognosis. The authors seem to be selecting for a cohort that is less likely to have aggressive features- which also will alter results, but in an opposite manner as presented by the authors already in the conclusion.
Response6: We are aware that it is a possible bias in our analysis: as kindly suggested, we specified in the discussion that we excluded the most aggressive forms from our analisis, thus representing a possible bias.

Reviewer 2 Report
Cimini and colleagues describe the evaluation of primary brain tumors with [18F]FDOPA and its correlation with IDH1 mutations and MGMTp methylation. The study addresses and important question and its very well executed. The study includes a large number of subjects and the methods are well described. The paper is well written.
The main finding is that there are no significant differences in tracer uptake in the tumor with IDH1 mutation and MGMTp methylation. Even though with negative findings it is not possible to rule out that in a larger and/or more homogeneous sample a statistically significant effect could be found, the results presented here are informative and support their conclusion.
This reviewer recommends acceptance after minor revision.
A few minor comments:
In the introduction please explain in more detail the mechanism of [18F]FDOPA
Throughout the manuscript, please use consistent notation of radiotracer names and follow nomenclature guidelines: “[18F]FDOPA” without spaces and the 18 superscript.
Methods: please mention how mutation status and methylation status were determined.
L123: standard “deviation”
L178: revise punctuation
Author Response
Point 1: In the introduction please explain in more detail the mechanism of [18F]FDOPA.
Response 1: As kindly suggested, we have explained the mechanism of [18F]FDOPA in the introduction, underlining its uptake mediated by LAT1 and 2 transporters (L 50-52). Moreover, we have specified that it is a L-DOPA analog (L 80-81), following the same metabolic pattern in cells.
Point 2: Throughout the manuscript, please use consistent notation of radiotracer names and follow nomenclature guidelines: “[18F]FDOPA” without spaces and the 18 superscript.
Response 2: As kindly suggested, we have written the correct form ([18F]FDOPA) throughout the manuscript.
Point 3: Methods: please mention how mutation status and methylation status were determined.
Response 3: As kindly suggested, we have mentioned how IDH1 mutation status and methylation status were detected (L 96).
Point 4: L123: standard “deviation”
Response 4: As kindly suggested, we have changed “device” to “deviation”.
Point 5: L178: revise punctuation
Response 5: As kindly suggested, we have revised punctuation.

Round 2
Reviewer 1 Report
The authors present a revised manuscript "MGMT promoter methylation and IDH1 mutations do not affect [18F]FDOPA uptake in primary brain tumors.
The authors have sufficiently addressed a majority of the comments made in the review, and it is understood that the additional requested information is not available due to limitations of the study design. The authors have included additional verbiage regarding the limitations of the study in the manuscript, which is helpful.
The entire paper (especially the introduction) would still benefit from review and revisions by someone proficient in the English language. Otherwise there are no further comments to the authors.
Author Response
Response to Reviewer 1 Comments
The entire paper (especially the introduction) would still benefit from review and revisions by someone proficient in the English language. Otherwise there are no further comments to the authors.
Response: As kindly suggested, the new version of the manuscript has been revised by a native English speaker. Several changes have been made, especially in the introduction.